



# General circulation models simulate negative liquid water path–droplet number correlations, but anthropogenic aerosols still increase simulated liquid water path

Johannes Mülmenstädt[1], Edward Gryspeerdt[2], Sudhakar Dipu[3], Johannes Quaas[3], Andrew S. Ackerman[4], Ann M. Fridlind[4], Florian Tornow[4,5], Susanne E. Bauer[4], Andrew Gettelman[1], Yi Ming[6], Youtong Zheng[7,8], Po-Lun Ma[1], Hailong Wang[1], Kai Zhang[1], Matthew W. Christensen[1], Adam C. Varble[1], L. Ruby Leung[1], Xiaohong Liu[9], David Neubauer[10], Daniel G. Partridge[11], Philip Stier[12], and Toshihiko Takemura[13]

[1]Atmospheric, Climate and Earth Sciences Division, Pacific Northwest National Laboratory, Richland, WA, USA
[2]Grantham Institute - Climate Change and the Environment, Imperial College London, UK
[3]Leipzig Institute for Meteorology, Leipzig University, Leipzig, Germany
[4]NASA Goddard Institute for Space Studies, New York, NY, USA
[5]Columbia University Center for Climate System Research, New York, NY, USA
[6]Boston College, Boston, MA, USA
[7]Atmospheric and Oceanic Science Program, Princeton University, Princeton, NJ, USA
[8]Department of Earth and Atmospheric Science, University of Houston, Houston, TX, USA
[9]Texas A&M University, College Station, TX, USA
[10]Institute for Atmospheric and Climate Science, ETH Zürich, Zurich, Switzerland
[11]Department of Mathematics and Statistics, University of Exeter, UK
[12]Atmospheric, Oceanic and Planetary Physics, Department of Physics, University of Oxford, UK
[13]Research Institute for Applied Mechanics, Kyushu University, Fukuoka, Japan

**Correspondence:** J. Mülmenstädt (johannes.muelmenstaedt@pnnl.gov)

**Abstract.** General circulation models' (GCMs) estimates of the liquid water path adjustment to anthropogenic aerosol emissions differ in sign from other lines of evidence. This reduces confidence in estimates of the effective radiative forcing of the climate by aerosol–cloud interactions (ERFaci). The discrepancy is thought to stem in part from GCMs' inability to represent the turbulence–microphysics interactions in cloud-top entrainment, a mechanism that leads to a reduction in liquid water in response to an anthropogenic increase in aerosols. In the real atmosphere, enhanced cloud-top entrainment is thought to be the dominant adjustment mechanism for liquid water path, weakening the overall ERFaci. We show that the latest generation of GCMs includes models that produce a negative correlation between present-day cloud droplet number and liquid water path, a key piece of observational evidence supporting liquid water path reduction by anthropogenic aerosols and one that earlier-generation GCMs could not reproduce. However, even in GCMs with this negative correlation, the increase in anthropogenic aerosols from preindustrial to present-day values still leads to an increase in simulated liquid water path due to the parameterized precipitation-suppression mechanism. This adds to the evidence that correlations in the present-day climate are not necessarily causal. We investigate sources of confounding to explain the noncausal correlation between liquid water path and droplet number. These results are a reminder that assessments of climate parameters based on multiple lines of evidence must carefully consider the complementary strengths of different lines when the lines disagree.





## 1 Introduction

Aerosol–cloud interactions (ACI) remain the greatest source of uncertainty in our estimates of anthropogenic perturbations to Earth's energy budget (Boucher et al., 2014; Forster et al., 2021). In liquid clouds, an anthropogenic aerosol perturbation essentially instantaneously alters the number of cloud droplets ($N_d$), changing cloud reflectance and thus the shortwave
radiation absorbed by the climate system, which exerts a radiative forcing on climate ("radiative forcing by aerosol–cloud interactions" or RFaci; Twomey, 1977; Boucher et al., 2014). While our knowledge of RFaci is uncertain (Quaas et al., 2020), an even thornier issue is cloud adjustments to the $N_d$ perturbation, where multiple processes acting at different scales from cloud droplet to planetary circulation (Stevens and Feingold, 2009) result in a multiscale dynamics prediction problem that is impervious to any one "silver bullet" solution (Mülmenstädt and Feingold, 2018). Estimates of ACI adjustments are, therefore,
based on multiple, and often conflicting, lines of evidence (Boucher et al., 2014; Bellouin et al., 2020; Forster et al., 2021). Those lines of evidence are, broadly, modeling at the cloud process scale ("large eddy simulation" or LES), global modeling, and observations at different scales.

In the following, we focus on stratocumulus (Sc) clouds, which play a large role in the energy budget due to their high albedo and frequent occurrence. Our understanding of adjustments in Sc is that two effects compete: an anthropogenic increase
in $N_d$ suppresses precipitation (Albrecht, 1989), increasing cloud liquid water path ($\mathcal{L}$); but the $N_d$ increase also promotes increasing turbulent entrainment of subsaturated air at cloud top (Ackerman et al., 2004; Bretherton et al., 2007), decreasing $\mathcal{L}$. These mechanisms are regime dependent; precipitation suppression only plays a role in clouds that would have precipitated in the absence of the aerosol perturbation, and the entrainment mechanisms depend strongly on the turbulence generation mechanisms, for example cloud-top radiative cooling. The regime dependence of the underlying processes leads to "process
fingerprints" in $N_d$–$\mathcal{L}$ space in LES (Hoffmann et al., 2020) for the very limited set of boundary conditions where LES is available. Similar bifurcation behavior appears in satellite observations, where mean $\mathcal{L}$ as a function of $N_d$ first increases in precipitating clouds, next reaches a peak that roughly coincides with the transition to nonprecipitating clouds, and then decreases again (Gryspeerdt et al., 2019). There is evidence that this "inverted v" relationship between $\mathcal{L}$ and $N_d$ overestimates the strength of the causal effect of $N_d$ on $\mathcal{L}$ (Gryspeerdt et al., 2019; Arola et al., 2022; Fons et al., 2023), but qualitatively it is
consistent with process understanding from LES. Integrated over all meteorological boundary conditions, the overall satellite correlation between $\mathcal{L}$ and $N_d$ is negative. The satellite inverted v, satellite observations of natural laboratories (Christensen et al., 2022) where the origin of the perturbation is evident (Malavelle et al., 2017; Toll et al., 2019), and process-modeling lines of evidence lead to the assessment that the adjustment of $\mathcal{L}$ to anthropogenic aerosol is a reduction of $\mathcal{L}$, that is, a positive contribution to the effective radiative forcing by ACI (ERFaci; Bellouin et al., 2020; Forster et al., 2021).
Global climate models – which, currently, means general circulation models (GCMs) run at roughly 1° latitude–longitude spatial resolution – tell a different story. They would project an increase, rather than a decrease, in $\mathcal{L}$ when aerosols are increased from preindustrial (PI) to present-day (PD) concentrations (Gryspeerdt et al., 2020). The GCM line of evidence is





discounted in multiline assessments because it conflicts with the other lines and because those lines are assumed to provide more reliable information. This assumption rests on the representation of the relevant processes in GCMs. In these models,

precipitation is initiated by a microphysical parameterization with an explicit dependence on $N_d$ (or, largely equivalent, droplet size), so that the $\mathcal{L}$ increase by precipitation suppression is explicitly parameterized. Reduced $\mathcal{L}$ by enhanced evaporation, on the other hand, depends critically on meter-scale or smaller interactions between turbulence, radiation, and microphysics at the cloud edge. These interactions fall between several parameterizations and are therefore tricky to formulate in GCMs. (As a perverse consequence, this causes us to fret that GCMs may be structurally incapable of representing turbulent entrainment

scales, while we often mistakenly consider the many-orders-of-magnitude-smaller-scale precipitation processes a parametric problem; e.g., Mülmenstädt et al., 2020, 2021).

In this work, we show that some Coupled Model Intercomparison Project 6 (CMIP6) era GCMs, unlike earlier model generations, are capable of producing inverted v $N_d$–$\mathcal{L}$ relationships in agreement with global observations and LES. Based on these PD correlations and on the $N_d$ change between PI and PD (i.e., mimicking the information available to observations-

based ERFaci estimates), these models predict a reduction in $\mathcal{L}$, which is consistent with assessments that use multiple lines of evidence. However, the causal effect of anthropogenic $N_d$ changes on $\mathcal{L}$, as diagnosed by model experiments where all climatic boundary conditions apart from aerosols are held fixed, remains as in previous GCM generations: an anthropogenic $N_d$ increase leads to an increase in average $\mathcal{L}$, consistent with a dominant role for the precipitation suppression mechanism parameterized in the model microphysics.

## 2  Data and methods

We use an ensemble of GCMs to perform fixed-sea surface temperature model experiments with PD and PI emissions, archive instantaneous aerosol and cloud information with sufficient frequency (3 h) to resolve the diurnal cycle and with sufficient length (1–5 years with the large-scale winds nudged to PD meteorology) to draw statistically robust conclusions. The model ensembles used are the CMIP5-era AeroCom indirect effect experiment (AeroCom IND3) simulations on the one hand and

four newer-version models prepared for CMIP6 on the other. The AeroCom models are described in Zhang et al. (2016); Ghan et al. (2016). The CMIP6-era models are the U.S. Department of Energy Exascale Earth System Model (E3SM) version 1 Atmosphere Model (EAMv1; Rasch et al., 2019), the NASA Goddard Institute for Space Studies (GISS) ModelE3 (Cesana et al., 2019, 2021) configuration Tun1, the Geophysical Fluid Dynamics Laboratory (GFDL) Atmospheric Model AM4.0 (Zhao et al., 2018), and the Community Earth System Model version 2 Community Atmosphere Model version 6 (CESM2-CAM6;

Gettelman et al., 2019). The CMIP6-era models were run for one year for the baseline experiment. E3SM was further run for 5 years for additional experiments that needed more data to perform stratification by confounding variables (see Sect. 3.3). For E3SM, the Cloud Feedback Model Intercomparison Project ObservaTon Simulator Package (COSP) satellite simulator (Pincus et al., 2012; Swales et al., 2018) mimicking the MODerate Resolution Imaging Spectroradiometer (MODIS) cloud retrievals (Platnick et al., 2017) and a number of vertically resolved fields were archived over a limited area over the northeast Pacific

(NEP) Sc region for further analysis of confounders.





## 2.1 Cloud selection

From the model output, we select liquid clouds, defined by the absence of ice (ice water path $< 10^{-3}$ kg m$^{-2}$ and ice cloud cover = 0) in the column. To mimic passive satellite analyses, as well as to simplify the application of entrainment diagnostics in part 2 of the series (Mülmenstädt et al., in prep.), we require near-overcast (liquid cloud cover > 0.9) conditions. For these
liquid clouds, we calculate "in-cloud" cloud-top $N_d$ and $\mathcal{L}$ by dividing the grid-mean $N_d$ and $\mathcal{L}$ by the projected cloud cover. Only clouds over ocean are considered in this analysis. We refer to these clouds as "overcast clouds".

In addition to these globally occurring overcast clouds, we also study smaller cloud subsets defined by dynamical regime following Medeiros and Stevens (2011). In this classification, the stratocumulus regime is based on vertical velocity $\omega$ at 700 and 500 hPa ($\omega_{700} > 10$ hPa d$^{-1}$ and $\omega_{500} > 10$ hPa d$^{-1}$) and lower tropospheric stability (LTS), which we define here as the
difference in potential temperature $\theta$ between 1000 hPa and 700 hPa ($\theta_{700} - \theta_{1000} > 18.55$ K). We further restrict the clouds to occur in grid boxes where these conditions are met at least 30% of the time, which serves to select the subtropical Sc regions. The occurrence fraction $f_{\text{Sc}}$ of these conditions is shown in Fig. 1. In addition to the Medeiros and Stevens (2011) requirements, all of the above-mentioned warm cloud criteria are applied. We refer to these clouds as "Sc regime clouds".

## 2.2 Analysis methods

From $\mathcal{L}$ and $N_d$, we construct the conditional probability $P(\mathcal{L}|N_d)$ following Gryspeerdt et al. (2019). For ease of comparison among models and configurations, we collapse the two-dimensional $P(\mathcal{L}|N_d)$ into one dimension by calculating the geometric-mean $\mathcal{L}$ in each $N_d$ bin, also following Gryspeerdt et al. (2019).

For the MODIS simulator analysis in Sect. 3.3.3, we transform the simulated $\tau$ and droplet effective radius ($r_e$) into $N_d$ and $\mathcal{L}$ using a power-law relationship for adiabatic updrafts with constant $N_d$ (Brenguier et al., 2000; Bennartz, 2007; Painemal
and Zuidema, 2011; Grosvenor et al., 2018):

$$N_d = \frac{\sqrt{5}}{2\pi k \sqrt{\rho_w Q}} \sqrt{f_{\text{ad}} \Gamma} \tau^{1/2} r_e^{-5/2} \tag{1}$$

$$\mathcal{L} = \frac{5}{9} \rho_w \tau r_e, \tag{2}$$

where we take the ratio $k = (r_v/r_e)^3$ between volumetric mean radius $r_v$ cubed and effective radius cubed to be 1, subadiabatic factor $f_{\text{ad}} = 1$, scattering efficiency $Q = 2$, and adiabatic condensation rate $\Gamma = 2 \times 10^{-6}$ kg m$^{-4}$. These assumptions minimize
the complications involved in showing results that are mostly power-law behavior independent of these constant factors. (This does neglect important modifications that can arise if these factors are not, in fact, constant; Varble et al., 2023).

To analyze confounding by planetary boundary layer (PBL) depth (Sect. 3.3.2), we identify the top of the Sc-like boundary layer by the first model level where temperature increases with height in Sc-regime overcast columns. This produces well-mixed profiles of liquid-water potential temperature $\theta_l$ and total water mixing ratio $q_w$. (Other definitions of PBL top, i.e.,
the model level of greatest gradient in $\theta_l$ or $q_w$, yield very similar results.) As we will see in Sect. 3.3.2, cloud and aerosol properties are remarkably stratified by PBL depth in E3SM; to keep the properties as distinct as possible as a function of



PBL depth, we retain the native model vertical discretization instead of converting the hybrid pressure levels to pressure or geometric height.

Table 1 summarizes the emissions, cloud selection, and model run duration for each experiment.

## 3 Results and discussion

In Fig. 2, we show the behavior of the AeroCom IND3 (CMIP5-era) GCMs in $N_d$–$\mathcal{L}$ space: with the exception of one model, $\mathcal{L}$ increases monotonically as a function of $N_d$. In some models, the slope decreases at high $N_d$, but only one model (HadGEM) has quantitatively similar behavior to the inverted v satellite $N_d$–$\mathcal{L}$ plot. The behavior of these models (with the exception of HadGEM) is consistent with the interpretation that the predominant mechanism linking $\mathcal{L}$ and $N_d$ is precipitation suppression.

### 3.1 CMIP6-era models produce inverted v $N_d$–$\mathcal{L}$ relationships

A funny thing happened on the way to CMIP6: three of the four US CMIP6-era GCMs have an inverted v with a pronounced negative slope. The behavior of these models is contrasted with the AeroCom models' behavior in Fig. 3. The geographic distribution of the regression slope between $\log\mathcal{L}$ and $\log N_d$ is predominantly negative in the models with an inverted v (Fig. 4), as Gryspeerdt et al. (2019) found in satellite retrievals.

One of these models (ModelE) was designed to better represent the entrainment behavior to which the negative slope is attributed in process-scale modeling. The other two (CAM6 and EAMv1), however, were not; if the negative slope is due to an entrainment ACI mechanism, it is an emergent behavior not explicitly parameterized into the turbulence scheme. It is doubly surprising that these models produce a negative slope considering that their closely related predecessor, CAM5.3-CLUBB-MG2, was part of the AeroCom ensemble and showed, at best, a slightly negative relationship between $N_d$ and $\mathcal{L}$.

### 3.2 The negative correlation between $N_d$ and $\mathcal{L}$ does not predict the sign of PI to PD change in $\mathcal{L}$

The bulk of the $N_d$ population lies in the part of the inverted v with a negative $N_d$–$\mathcal{L}$ correlation. If we regarded this relationship as indicative of a causal influence of $N_d$ on $\mathcal{L}$ – that is, that an increase in $N_d$ causes $\mathcal{L}$ to decrease – then we would predict a decrease in $\mathcal{L}$ as $N_d$ increases from its PI value to its PD value due to anthropogenic emissions.

We can compare the change in $\mathcal{L}$ predicted by the $N_d$–$\mathcal{L}$ correlation in PD internal variability to the outcome of a model experiment designed to measure the causal effect of $N_d$ on $\mathcal{L}$. This experiment fixes all climatic boundary conditions affecting cloud state (i.e., solar constant, greenhouse gases, and sea-surface temperature) with the exception of anthropogenic aerosols. The change in $\mathcal{L}$ in this experiment can therefore only be due to the anthropogenic aerosol emissions change. This model experiment shows that the causal effect of the $N_d$ increase is to increase $\mathcal{L}$ on average, contradicting the prediction of a decrease in $\mathcal{L}$ based on PD internal variability (Fig. 5). The correlation seen in PD internal variability in these models therefore cannot be causal. Plotting the correlations within PD and PI, as shown in Fig. 6, provides a glimpse at what is happening instead: a secular increase in $N_d$ does not lead to a secular reduction in $\mathcal{L}$ by shifting the $\mathcal{L}$ population along the correlation line, as would be expected for a causal relationship. Instead, the correlation line shifts along with the secular shifts in $N_d$ and





$\mathcal{L}$ ( mostly to the right given that the change in $N_d$ is far greater than the change in $\mathcal{L}$) in a way that is not predicted by the correlation line itself.

This contradiction raises three questions. First, what produces the noncausal negative $N_d$–$\mathcal{L}$ correlation? We provide a few hypotheses in the following section. Second, considering that these models can replicate the observed PD correlation, what can we infer about the causality of the relationship in observations, where we are unable to conduct direct experimental tests of causality? We discuss this question in Sect. 3.4. Third, is any part of the negative relationship between $N_d$ and $\mathcal{L}$ in the models causal? Any such causal mechanism would have to involve a direct or indirect $N_d$-dependence in cloud-top
entrainment. In ModelE, the Bretherton and Park (2009) turbulence scheme provides an explicit entrainment closure. Guo et al. (2011) have shown that the combination of the Cloud Layers Unified By Binormals (CLUBB; Larson and Golaz, 2005; Golaz et al., 2007) cloud and turbulence scheme and the Morrison–Gettelman microphysics scheme (Morrison and Gettelman, 2008; Salzmann et al., 2010) can reproduce entrainment-mediated enhanced evaporation at high $N_d$ in single-column experiments. This behavior has not been documented in three-dimensional GCM experiments, but CAM6 and EAMv1 use related cloud–
turbulence (Bogenschutz et al., 2013; Larson, 2022) and cloud–microphysics (Gettelman, 2015) schemes, so it is conceivable that $N_d$-dependent entrainment mechanisms contribute to the $N_d$–$\mathcal{L}$ relationship in these three models. A deeper investigation of this question merits a separate paper (part 2 of this series, Mülmenstädt et al., in prep.).

### 3.3    Sources of covariability that produce noncausal $N_d$–$\mathcal{L}$ relationships

Noncausal relationships between two variables often originate from a third (possibly unobserved) variable that exerts a causal
relationship on the two variables being correlated. This third variable is termed a "confounding variable" (Pearl and Mackenzie, 2018). In its most striking form, confounding can lead to a sign reversal between causation and correlation, for example in Simpson's paradox (Simpson, 1951; Feingold et al., 2022). Cloud properties respond strongly to the circulation at the scales of the Sc cellular organization (mesoscale) and greater. Thus, the meso- to synoptic-scale circulation is a natural place to look for confounding variables that lead to noncausal correlations between cloud properties.

#### 3.3.1    Mesoscale cloud regimes

Mesoscale circulation manifests as cloud "regimes" (e.g., Rossow et al., 2005; Gryspeerdt and Stier, 2012; Muhlbauer et al., 2014; Unglaub et al., 2020). ACI mechanisms likely differ between cloud regimes (e.g., Mülmenstädt and Feingold, 2018; Possner et al., 2020; Dipu et al., 2022). This could result in different $N_d$–$\mathcal{L}$ slopes in open- or closed-cell Sc or shallow cumulus or, as the positive- and negative-sloped legs of the inverted v relationship perhaps show, in precipitating and nonprecipitating
cloud regimes. Due to GCMs' coarse resolution, it is doubtful that they can correctly represent these mesoscale cloud regimes, their ACI mechanisms, or their coupling to the circulation. Nevertheless – or perhaps precisely because we can probably discount cloud-scale causal links between $N_d$ and $\mathcal{L}$ due to the mismatch with the GCM resolved scale – we can use GCMs to test whether the existence of cloud regimes is, on its own, a confounding mechanism for the $N_d$–$\mathcal{L}$ relationship.

To assess whether regime-induced confounding effects may exist in the model $N_d$–$\mathcal{L}$ relationship, we stratify the E3SM
model clouds by surface rain rate. These bins of rain rate are our stand-in for precipitation regimes. We focus on the surface





rain rate because, unlike mesoscale morphological regime definitions (which are subgrid scale in the GCM), the precipitating–nonprecipitating regime delineation has a somewhat clear analog in the GCM. Because the model rain rate has a very long low tail, we do not attempt to define a binary nonprecipitating versus precipitating categorization but rather divide the cloud sample into quantiles of rain rate. Specifically, we use sextiles, balancing the need for a meaningful range of rain rates with the
need to maintain a large sample of clouds within each bin. The CloudSat precipitation detection sensitivity at the GCM spatial resolution ($\approx 0.01$ mm d$^{-1}$; Stephens et al., 2010) falls roughly into the third rain rate bin, so, by this definition, half the bins approximately represent precipitating and half nonprecipitating clouds.

Figure 7 shows the results. The model, perhaps unrealistically, produces clouds that generate surface-reaching rain at all droplet concentrations; however, in bins with higher rain rates, the $N_d$ distribution is noticeably lower, as might be expected
from the negative-exponent power law that parameterizes the autoconversion of cloud water to rain, and as is expected from observations (Pawlowska and Brenguier, 2003; Comstock et al., 2004). At the same time, $\mathcal{L}$ is higher in bins with higher rain rates, again as might be expected from the parameterized autoconversion and accretion. Superimposing the bin-mean $N_d$ and $\mathcal{L}$ for each rain-rate bin on the unbinned $N_d$–$\mathcal{L}$ distribution, we find that the negative correlation among the bin means echoes the unbinned correlation. This is the case even though, in very classic Simpson (1951) fashion, the correlations within five
out of the six $R$ bins are positive. Thus, the opposing influences of $N_d$ and $\mathcal{L}$ on rain rate can, without any involvement of entrainment or evaporation mechanisms, generate a noncausal negative correlation between $N_d$ and $\mathcal{L}$.

We note that the mechanism generating this noncausal correlation is unusual. The contrast with the causal precipitation suppression mechanism is clear: there, causation runs from $N_d$ to autoconversion to $\mathcal{L}$. Here, causation runs jointly from both $N_d$ and $\mathcal{L}$ to precipitation. How or whether the causal chain then returns from precipitation to $N_d$ and $\mathcal{L}$, as in the classic
confounding mechanism, is an open question.

We further note that precipitation already appears to have a qualitative effect on the model's $N_d$–$\mathcal{L}$ relationship at rain rates far below the CloudSat sensitivity threshold: even in the second-lowest $R$ bin, the correlation between $N_d$ and $\mathcal{L}$ is already positive. This suggests that the parameterized precipitation may exert such a strong influence on ACI even for clouds with low precipitation rate that other ACI adjustment mechanisms, while they may in principle be represented in the model, could be so
overwhelmed by the parameterized precipitation suppression that their effect is not discernible in the climate response.

### 3.3.2 Synoptic-scale airmass advection

At the synoptic to planetary scales, covariability between cloud and aerosol properties can lead to spurious correlations in ACI metrics (Grandey and Stier, 2010). Synoptic-scale meteorological covariability can take the form of continental versus marine airmass advection. When an airmass originates over land, it typically has higher temperature, lower relative humidity
(contributing to lower $\mathcal{L}$), and higher aerosol concentration (contributing to higher $N_d$) than when an airmass originates over ocean. This contrast between airmasses creates an anticorrelation between $N_d$ and $\mathcal{L}$ even in the absence of any causal effect of $N_d$ on $\mathcal{L}$ (Brenguier et al., 2003). Additionally, sea surface temperature is coldest and climatological subsidence strongest, near the coast, resulting in shallow marine boundary layers. The model's conception of this synoptic-scale covariability in space can be seen in Fig. 8, with shallow boundary layers and high cloud condensation nuclei (CCN) concentrations near shore and



deeper boundary layers with low CCN farther offshore. A similar covariability exists at particular locations in time. Figure 9 illustrates the mechanism in the NEP Sc region: at any given location, PBL depth and CCN concentration are strongly linked via the synoptic-scale circulation. Presumably the position of the anticyclonic subtropical subsidence governs both the PBL depth and whether continental or maritime air is advected.

To assess the synoptic meteorological confounding effect, we stratify the E3SM model clouds in the NEP Sc region by PBL
depth. We choose PBL depth as the confounding variable because it appears to act as a proxy for airmass "continentality" in the model (Fig. 8), without a direct parameterized relationship to either aerosols or cloud. PBL depth is nevertheless strongly correlated with both CCN concentration (temporal- and regional-mean vertical profiles are shown in Fig. 10) and $\mathcal{L}$. For a fairly wide range of PBL depths (representing the central 90% of the PBL depth distribution for Sc-regime cloud columns), the relationship between mean $N_d$ and mean $\mathcal{L}$ stratified by PBL depth mimics the slope of the unstratified $N_d$–$\mathcal{L}$ relationship
quite closely (Fig. 11). Based on this, it is plausible that synoptic-scale meteorological covariability contributes substantially to the overall negative $N_d$–$\mathcal{L}$ correlation in the model.

We note several caveats. The synoptic-scale covariability of aerosol advection and PBL depth is a feature of the general circulation and can therefore be expected to be modeled reasonably well in a GCM. However, the interaction of the advected aerosol with boundary-layer clouds depends on mixing between the free troposphere and the boundary layer, which is likely
much less well represented in GCMs that have coarse vertical resolution. Whether the synoptic-scale confounding signature in the model mimics the real atmosphere is therefore uncertain. Further, the synoptic-scale covariability differs depending on the geographic particulars of each Sc basin; we have only analyzed the NEP Sc in detail. Finally, while the PBL depth-stratified negative $N_d$–$\mathcal{L}$ relationship in the model is consistent with observational analyses (e.g., Fons et al., 2023), the model does not reproduce the weakening of the $N_d$–$\mathcal{L}$ correlation within each PBL depth bin (not shown) that is found in observations
(Possner et al., 2020; Fons et al., 2023).

### 3.3.3   Phase-space boundaries

Correlations between $N_d$ and $\mathcal{L}$ can also arise simply because not all parts of the $N_d$–$\mathcal{L}$ phase space are equally accessible to clouds. This can be illustrated by applying the MODIS simulator (Pincus et al., 2012) to the model. The MODIS simulator provides optical thickness $\tau$ and droplet effective radius $r_e$ diagnosed consistently with the MODIS satellite cloud retrievals
(Platnick et al., 2017). Power-law adiabatic relationships $\mathcal{L}(r_e, \tau)$ and $N_d(r_e, \tau)$ can be used to transform the MODIS output into $N_d$–$\mathcal{L}$ space (e.g., Dipu et al., 2022). In logarithmic coordinates, this is a linear transformation, yielding the correlation shown in Fig. 12. This S-curve correlation, like the model-native $N_d$–$\mathcal{L}$ correlation, shows a steep rise in $\mathcal{L}$ at low $N_d$ and a steep drop at moderate $N_d$. It also shows another steep rise at high $N_d$ that the model may hint at but does not exhibit clearly. Investigating the data before and after the coordinate transformation to $N_d$–$\mathcal{L}$ space is instructive. In $\log \tau$–$\log r_e$ space, the
MODIS simulator output falls within a rectangle, bounded by the limits the model prescribes on its clouds. Upon transformation to $\log \mathcal{L}$–$\log N_d$ space, the population is bounded by a parallelogram (see the isolines in Fig. 12). These limits on the phase space strongly sculpt the behavior of the mean $\log \mathcal{L}$ as a function of $\log N_d$, because the parts of phase space that are not populated do not contribute to the mean $\mathcal{L}$ as a function of $N_d$.





### 3.4 Persistent disagreement with other lines of evidence

Before these results, it was only logical to discount the GCM evidence on the basis that it could not reproduce the observed the $N_d$−$\mathcal{L}$ relationship in PD internal variability. Now that some GCMs match the other lines of evidence in PD internal variability, what do we make of the fact that the disagreement on the sign of the causal climatic $\mathcal{L}$ adjustment to RFaci persists?

In observations, it is more difficult to establish causality than in the GCMs, where it is as simple as changing the aerosol emissions while fixing all other boundary conditions. The most reliable causal evidence in observations comes from observational

natural laboratories where the aerosol perturbation is known and an unperturbed control can be identified clearly (Christensen et al., 2022). Such laboratories indicate unchanged or reduced $\mathcal{L}$ in the perturbed clouds (Malavelle et al., 2017; Toll et al., 2019; Diamond et al., 2020). But such laboratories are rare, and there is no rigorous extrapolation from these laboratories to the full diversity of cloud regimes found in the climate. The most representative observations, that is, the global satellite-retrieved inverted v correlations, have the opposite problem: they are representative, but are the correlations causal? The correlation is

more negative than the estimate of causal interannual $\mathcal{L}$ response to $N_d$ perturbations using an effusive volcano as a laboratory (Gryspeerdt et al., 2019, albeit for shallow Cu rather than Sc). Arola et al. (2022) argue that satellite $N_d$−$\mathcal{L}$ correlations are negatively biased not only by covariability confounding but also by retrieval errors. Fons et al. (2023) applied a causal network approach to the temporal evolution in geostationary satellite data and found that the causal negative $N_d$−$\mathcal{L}$ relationship is weaker than the $N_d$−$\mathcal{L}$ correlation. Strong regional increasing and declining trends on multidecadal timescales in the satellite

record may also contribute to disentangling covariability and causality (Quaas et al., 2022).

In LES, as in GCMs, causality can be established by varying aerosol concentration while keeping the other boundary conditions constant. This provides very clear evidence that precipitation suppression and entrainment feedbacks lead to process fingerprints of positive and negative $\mathcal{L}$ tendencies in $N_d$−$\mathcal{L}$ space (Hoffmann et al., 2020) that translate into steady Sc states (Glassmeier et al., 2021). But these LES experiments are expensive, so boundary conditions are carefully curated to a very small

subset of the high-dimensional space of meteorological conditions present in the climate. We simply do not know whether the process fingerprints would be as unambiguous if a broader spectrum of boundary conditions were simulated or if the clouds were able to interact with larger scales in the multiscale climate problem (Kazil et al., 2021) instead of evolving to a steady state.

In summary, GCMs are still the odd ones out in their negative $\mathcal{L}$ adjustment component of ERFaci. The observational and

LES modeling lines of evidence have clear confounding and representativeness problems. Are these problems severe enough to flip the sign of the adjustment? It seems unlikely, but our GCM results show that it is possible; addressing the representativeness and confounder questions in the other lines of evidence thus takes on a renewed urgency.

### 4 Conclusions

Mülmenstädt and Wilcox (2021) expressed the hope that global models, after a long stretch of playing the odd line of evidence

out in assessments of global energy budget problems (Bellouin et al., 2020; Sherwood et al., 2020), might be returning to a more equal role in the balance and struggle between conflicting lines of evidence. One way in which the global model perspective



shores up the strength of the multiline assessment by providing information not available from the other lines of evidence: being able to test causality and showing that PD internal variability may not even correctly predict the sign of the causal cloud water adjustment to the anthropogenic cloud droplet perturbation.

Causality (or, in this case, lack of causality) is easy to establish in a model experiment but very difficult in observations. Where the noncausal correlation originates is another question that models can, in principle, answer definitively by shutting off confounding model mechanisms in mechanism-denial experiments. In part 2 of this series, we will more fully use the power of models as hypothesis testers by performing perturbed-physics and mechanism-denial experiments. In this paper, we have restricted ourselves to slice-and-dice analyses that could, in principle, also be performed on observations. We hope that

they will be performed on observations, especially if Lagrangian investigation of cloud life cycle (e.g., Eastman et al., 2022; Christensen et al., 2023) and observational fingerprints of loss processes (e.g., Varble et al., 2023) can be included.

Whether lack of causality in the model system implies lack of causality in the real atmosphere is a question that models alone cannot address, so we do not yet know how worried we need to be about the sign difference between correlation and causation in the model $N_d$–$\mathcal{L}$ relationship. When it comes to the non-GCM lines of evidence, one can quibble with the representativeness of the causal evidence and with causality in the representative evidence – at the very least, these model results are a flashing

red warning sign hanging over our interpretation of the $\mathcal{L}$ adjustment component of ERFaci.

*Code and data availability.* Following acceptance, the analysis code and model output will be released with a code and data DOI

*Author contributions.* All authors contributed to the experiment design, model runs, data analysis, or manuscript writing.

*Competing interests.* At least one of the (co-)authors is a member of the editorial board of Atmospheric Chemistry and Physics.

*Acknowledgements.* We thank Christopher Bretherton, Susannah Burrows, Yao-Sheng Chen, Leo Donner, Graham Feingold, Jan Kazil, Naser Mahfouz, Daniel McCoy, Isabel McCoy, Christina Sackmann, Rob Wood, Jianhao Zhang, and Xiaoli Zhou for comments and discussion. This work arises from the 2021 U.S. Climate Modeling Summit held virtually and cochaired by Susanne Bauer and Gokhan Danabasoglu. JM was supported by the Office of Science, U.S. Department of Energy (DOE) Biological and Environmental Research as part of the Regional and Global Model Analysis program area and used resources of the National Energy Research Scientific Computing Center

(NERSC), a U.S. DOE Office of Science User Facility located at Lawrence Berkeley National Laboratory, operated under contract DE-AC02-05CH11231. ASA, AMF, FT and SEB were supported by the NASA Modeling, Analysis, and Prediction Program and their computational resources were provided by the NASA Center for Climate Simulation (NCCS) at Goddard Space Flight Center. EG was supported by a Royal Society University Research Fellowship (URF\R1\191602). DN acknowledges support from the European Union's Horizon 2020 research and innovation program (grant agreement no. 821205. TT was supported by the Japan Society for the Promotion of Science (JSPS) KAK-



ENHI (JP19H05669) and the Environment Research and Technology Development Fund S-20 (JPMEERF21S12010) of the Environmental Restoration and Conservation Agency provided by the Ministry of the Environment, Japan. PS was supported by the European Research Council project RECAP under the European Union's Horizon 2020 research and innovation program (grant no. 724602) and the FORCeS project under the European Union's Horizon 2020 research and innovation program (grant no. 821205). The Pacific Northwest National Laboratory (PNNL) is operated for DOE by Battelle Memorial Institute under contract DE-AC05-76RLO1830.





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





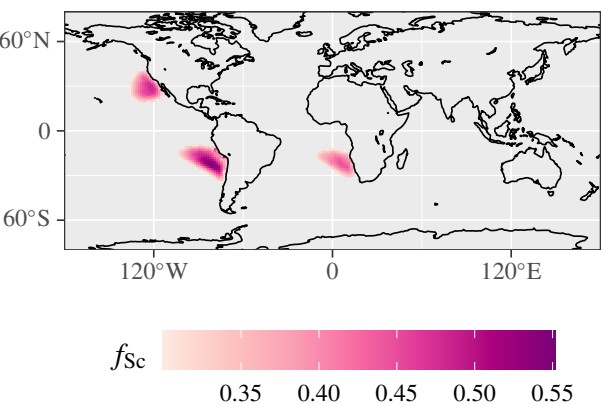

**Figure 1.** Occurrence fraction $f_{Sc}$ of Sc conditions by the Medeiros and Stevens (2011) criteria in EAMv1, shown where $f_{Sc} > 0.3$ (NEP, southeast Pacific, southeast Atlantic Sc regions).



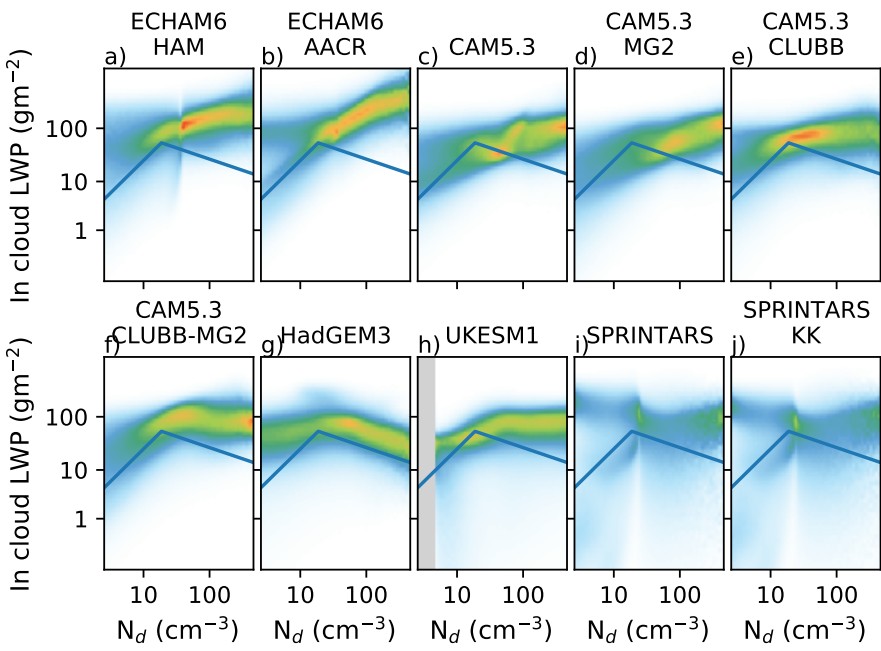

**Figure 2.** AeroCom IND3 state-of-the-art models' $N_d$–$\mathcal{L}$ relationship. The satellite inverted v relationship (Gryspeerdt et al., 2019) is indicated by the solid line.

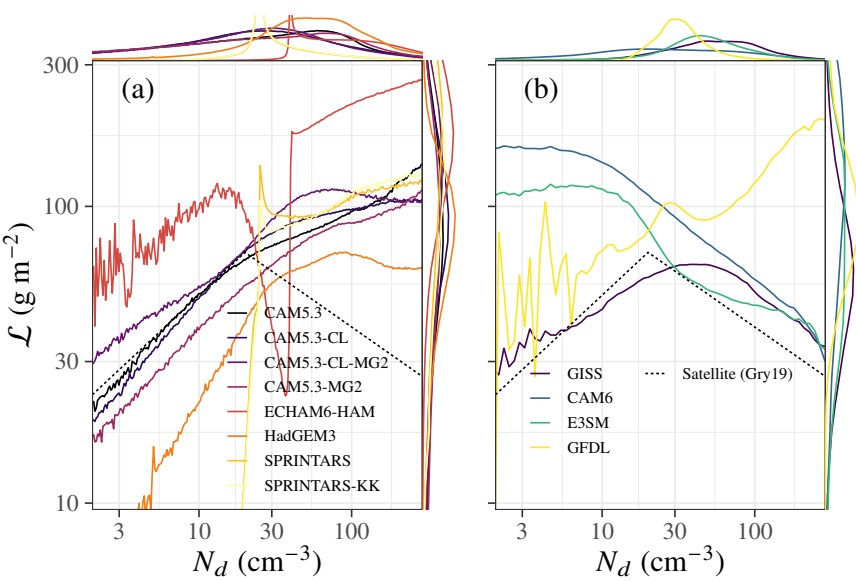

**Figure 3.** AeroCom IND3 state-of-the-art models' marginal distributions of $N_d$ and $\mathcal{L}$ and $N_d$–$\mathcal{L}$ relationship (a) compared to the CMIP6-era state-of-the-art models' $N_d$–$\mathcal{L}$ relationship (b). The satellite inverted v relationship (Gryspeerdt et al., 2019) is indicated by the dotted line. Three of the four CMIP6 models examined are qualitatively similar to the satellite result in the sense that the $N_d$–$\mathcal{L}$ correlation turns negative at moderate $N_d$.



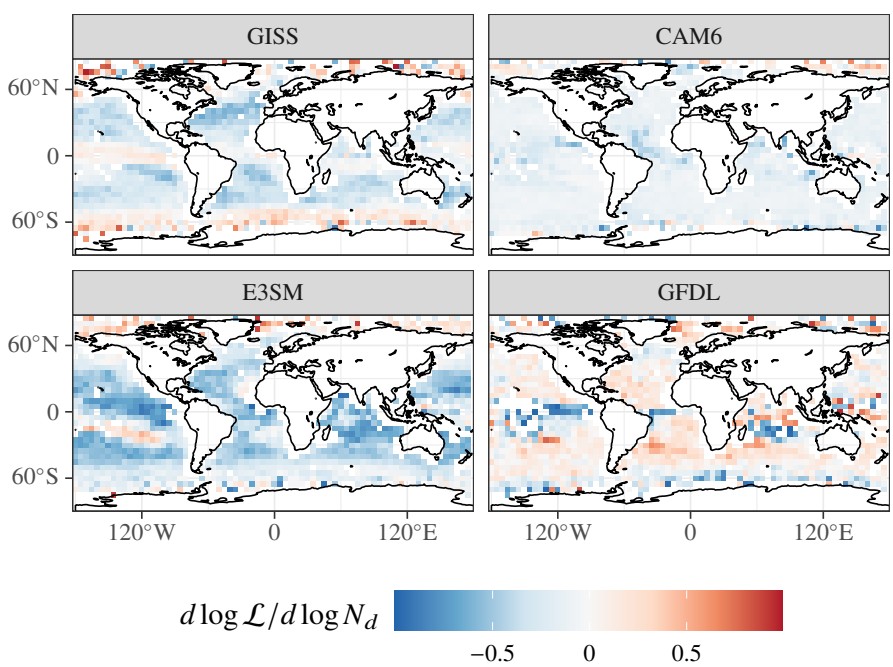

**Figure 4.** Geographic distribution of $d\log\mathcal{L}/d\log N_d$. Model output is aggregated to $5°\times5°$ latitude–longitude boxes before calculating linear regression slopes of $\log\mathcal{L}$ against $\log N_d$.





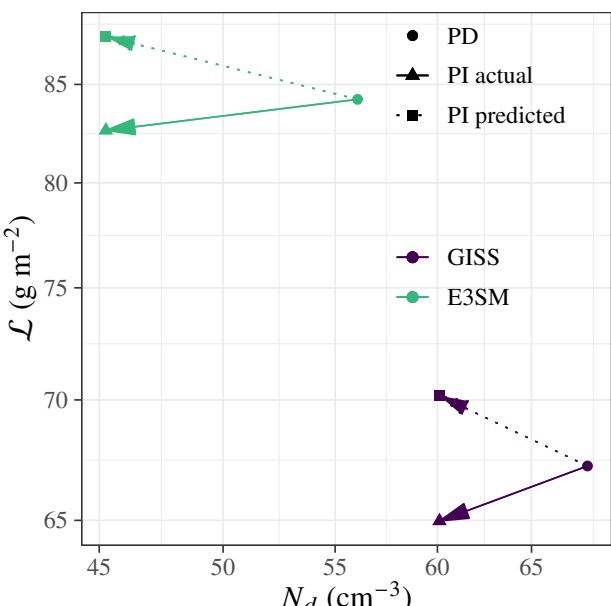

**Figure 5.** PI–PD $\mathcal{L}$ change from the causal experiment (solid arrow) contrasted with the change predicted from the PD internal variability (dashed arrow). The mean $\log \mathcal{L}$ as a function of $\log N_d$ from the PD model run (Fig. 3) is used to predict PI mean $\log \mathcal{L}$ from the PI $\log N_d$ distribution. Even though the PD $N_d$–$\mathcal{L}$ correlation is negative, $\mathcal{L}_{\mathrm{PD}} > \mathcal{L}_{\mathrm{PI}}$.



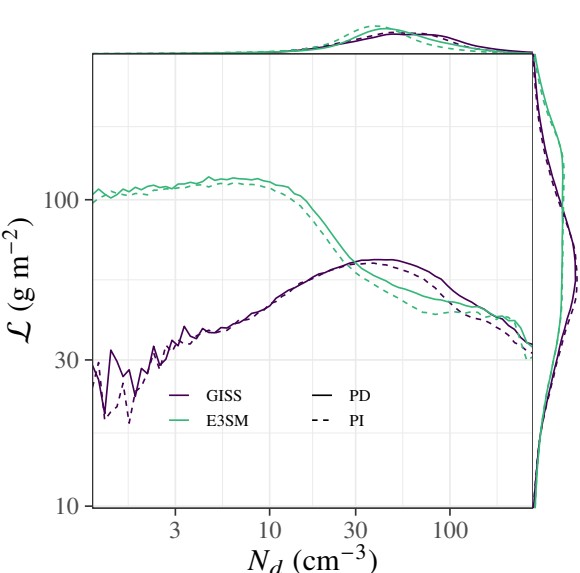

**Figure 6.** PD (solid) and PI (dashed) $N_d$ and $\mathcal{L}$ marginal distributions and $N_d$–$\mathcal{L}$ correlation in two GCMs with unrelated turbulence schemes. The $N_d$–$\mathcal{L}$ relationship based on internal variability within one climate state is not universal across the states.



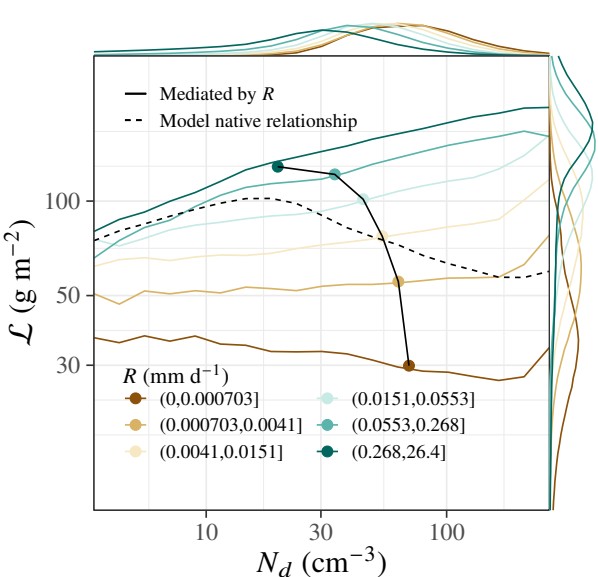

**Figure 7.** Precipitation-stratified $N_d$ and $\mathcal{L}$ marginal distributions and $N_d$–$\mathcal{L}$ relationships (colored lines). The dashed black line shows the unstratified $N_d$–$\mathcal{L}$ relationship. The solid black line connects the mean $(N_d, \mathcal{L})$ in each precipitation sextile (colored dots). Binning by precipitation intensity exposes a precipitation-mediated negative $N_d$–$\mathcal{L}$ covariability with a much steeper slope than the overall $N_d$–$\mathcal{L}$ correlation, even though the $N_d$–$\mathcal{L}$ correlation within all but the least precipitating sextile is positive.



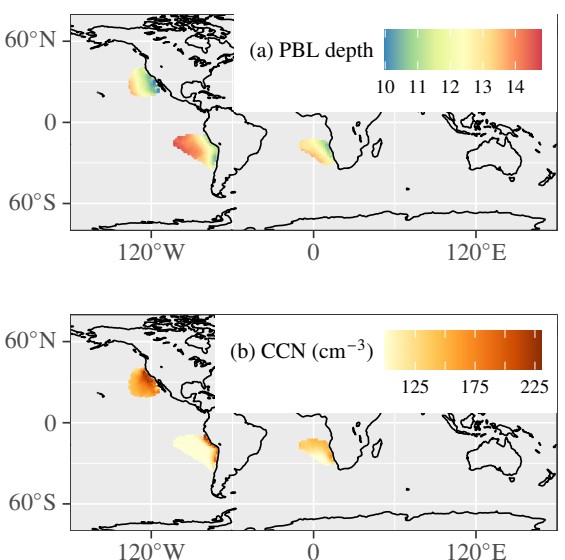

**Figure 8.** Within the Sc regime, synoptic-scale meteorology results in strong spatial covariability of (a) PBL depth (measured in model levels) and (b) CCN concentration at 0.2% supersaturation averaged over the depth of the PBL. Both of these variables are functions of airmass continentality.





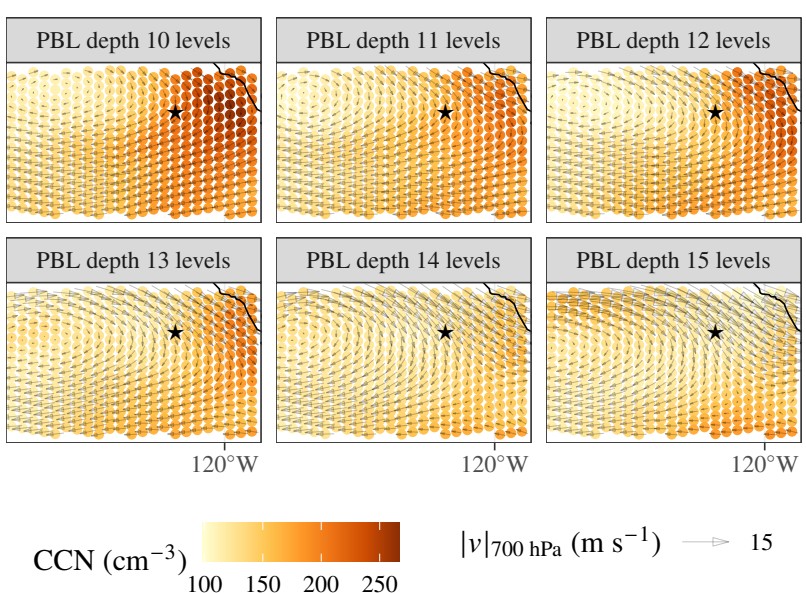

**Figure 9.** Within the Sc regime, synoptic-scale meteorology results in strong temporal covariability between CCN and PBL depth. This is exemplified by the NEP; the Southern California Bight is depicted in top right corner. The star indicates the grid point with the highest occurrence fraction of Sc conditions according to the criteria of Medeiros and Stevens (2011). (Aggregates exclude points over land.)

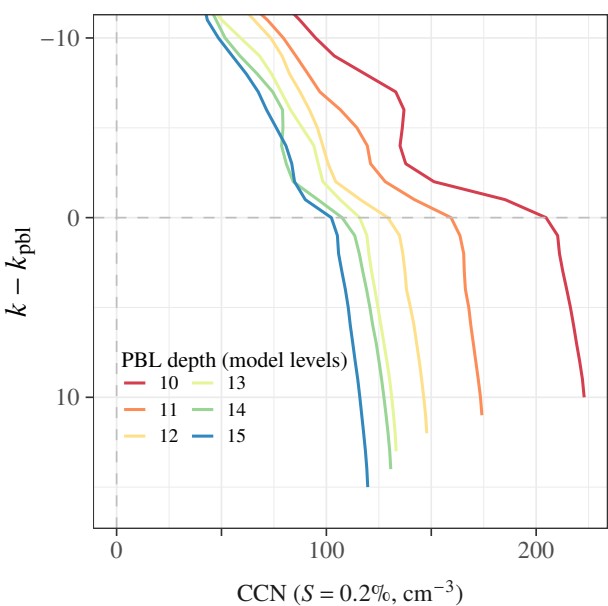

**Figure 10.** Temporal and regional-mean CCN concentration profiles in the NEP Sc region stratified by PBL depth. Within the Sc regime, CCN is strongly sorted by PBL depth, illustrating the strong covariability between PBL thermodynamic structure and aerosol advection. The central 90% of the PBL depth range (between 10 and 15 model levels, corresponding approximately to between 750–1400 m) is shown to avoid outliers in the low-statistics PBL depth bins. The PBL depth is measured in units of model levels $k$, with $k$ decreasing downward from the level of the PBL-capping inversion $k_{pbl}$ to the model level closest to the surface ($k = 72$ in EAMv1).



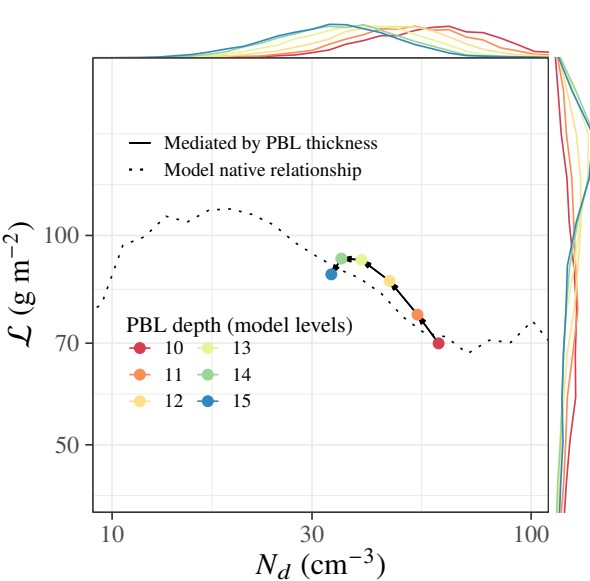

**Figure 11.** Within the Sc regime, PBL depth–CCN covariability leads to a negative $N_d$–$\mathcal{L}$ correlation with slope similar to the overall $N_d$–$\mathcal{L}$ correlation. The dashed black line shows the $N_d$–$\mathcal{L}$ relationship not stratified by PBL depth. The solid black line connects the mean $(N_d, \mathcal{L})$ at each PBL depth (colored dots). The central 90% of the PBL depth range (between 10 and 15 model levels) is shown to avoid outliers in the low-statistics PBL depth bins.



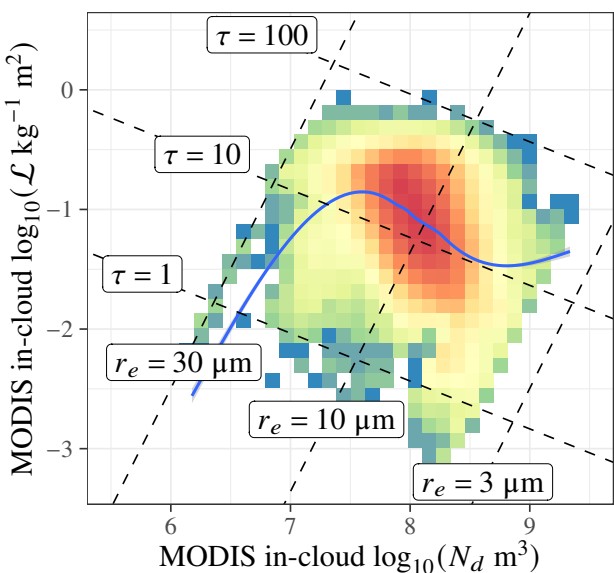

**Figure 12.** Joint probability distribution of E3SM MODIS-simulated $\mathcal{L}$ and $N_d$. Isolines of $r_e$ and $\tau$, from which adiabatic $N_d$ and $\mathcal{L}$ are retrieved, are overlaid. The mean $\log \mathcal{L}$ as a function of $\log N_d$ is shown as a blue line. Because the model imposes a rectangular limit on $r_e$ and $\tau$, the $N_d$–$\mathcal{L}$ phase space has parallelogram-shaped boundaries. At least part of the rise, fall, and repeated rise of $\mathcal{L}$ as a function of $N_d$ (blue line) is due to these phase-space boundaries cutting off the upper and lower parts of the $\mathcal{L}$ distribution.



**Table 1.** GCM experiments used in this analysis.

| Experiment | Emissions | Cloud selection | Duration |
|---|---|---|---|
| Multimodel PD | Present-day | Global overcast | 2010, nudged |
| Multimodel PI | Preindustrial | Global overcast | 2011, nudged |
| E3SM + COSP simulator | Present-day | NEP Sc regime | 2010, nudged |
| E3SM precip | Present-day | Sc regime | 2010–2014, nudged |