# Peer review of "General circulation models simulate negative liquid water path–droplet number correlations, but anthropogenic aerosols still increase simulated liquid water path"

_EGUsphere, 2024_

## Referee Comment (RC1)

**Review of "General circulation models simulate negative liquid water path–droplet number correlations, but anthropogenic aerosols still increase simulated liquid water path" by Johannes Mülmenstädt et al.**

This paper presents an interesting "seeming paradox" in the GCM realm, that is the latest generation of GCMs reproduced the inverted-v Nd-LWP relationship as has been observed by satellite-based studies, whereas PD-PI causal experiments using the same set of models produce the opposite response, which is consistent with the parameterized precipitation-suppression mechanism. It's not really a "paradox" in the GCM realm, as we know only the latter is causal. To figure out the causes of the inverted-v Nd-LWP relationship, the authors provided a thorough investigation on sources of confounding/covariability that evidently contribute to the noncausal Nd-LWP relationship manifested in GCMs' internal variability.

By unraveling the cause of the seeming paradox, this study concludes several key points with profound impacts on the ACI community:
1) Using the GCM framework with the capability to test causality, the authors show the possibility that correlations observed in PD mean climate states are not necessarily causal, and my not even represent the correct sign of causal LWP adjustment to aerosol perturbations.
2) When interpreting LWP adjustments from multiple lines of evidence, cautions are needed, and the causal piece of information should be taken into account and carefully weighted when integrating these lines of evidence.
3) There is pressing need to address the representativeness and confounder questions in non-GCM lines of evidence.

For me personally, this manuscript is extremely intriguing and has stimulated a lot of fruitful thinking on my end! The text is constructed in a clear, easy-to-follow and attractive storytelling manner. I enjoyed reading it, and I believe there is no doubt that this manuscript is worthy of speedy publication to raise community awareness of these impactful conclusions.

**Stimulated thoughts after reading (rather than comments):**

- Regarding the disagreement between the causal experiment (PD-PI) and the internal variability, is there a possibility that the causal experiment is missing some feedback mechanisms (at longer timescales) that may be present in the internal variability, because of the fixed climatic boundary conditions? (I realize this may not contribute much to the disagreement, but just wondering…) About the climatic boundary conditions, you mentioned that SST is fixed, is the circulation (winds) also fixed? (I'm not very familiar with the setup of these experiments)
  For now, let's assume the internal variability (inverted v) captures the mean climate state where MET (large-scale conditions), Nd, and LWP are in balance (manifested in some climate scale correlations), perturbing Nd initially causes changes in LWP, which may later lead to circulation and/or SST changes (feedback from LWP to MET, and then possibly back to LWP). Is this potential feedback pathway artificially shutoff in these PI-versus-PD runs, based on the configuration?

- Regarding the "funny" "doubly surprising" thing happened in CMIP6 models. I'm just curious is there any clue on what causes the CMIP6 models to get this inverted v (I understand the case for ModelE)? Are there any speculations? Is this due to the fact that the newer version of models better capture the mean climate states, thereby closer to observationally derived correlations? A following question is that if you use AeroCom IND3 models to predict the PI-LWP, would you get agreement with the causal experiment?

**Some notes:**

- Line 77, check spelling "ObservaTon"

- Figure 7-12, perhaps it's worth mentioning these are results from E3SM in the captions? (I know this is clearly indicated in the main text, so, feel free to ignore this).

- Figure 9, wind vectors are kind of hard to see, I suggest enlarging them (perhaps fewer of them will help too); is it better to indicate translated PBL depth in pressure or meter (more intuitive units)?

- Line 229 & Figure 11, regarding Nd-LWP correlation within each PBL depth bin (not shown), I wonder if it's worth showing, as I am curious about whether they look similar to what have been shown in Figure 7, i.e., in classic Simpson fashion, or different?

- *Just want to say that I really enjoyed reading Section 3.4 and the conclusion part. Great discussions! and I think the ACI community should really think carefully along these lines (i.e., representativeness versus/and causality) before producing tons of papers on the topic while not sure about how much of the results are causal.*

Looking forward to reading part 2 of the series!

Regards,

Jianhao Zhang

---

## Referee Comment (RC2)

Review of "General circulation models simulate negative liquid water path–droplet number correlations, but anthropogenic aerosols still increase simulated liquid water path"
by Johannes Mülmenstädt et al.

Summary:

The paper addresses strengths and weakness with respect to causation and representativeness of correlations of different strains of evidence assessing the liquid water path (LWP) adjustment due to changes in anthropogenic aerosol and consequently droplet number concentration ($N_d$). In particular, methods used to establish correlations between LWP and $N_d$ in present-day remote sensing climatology are applied to global climate model (GCM) simulations in four CMIP6 models of which three do indeed simulate a negative correlation between LWP and $N_d$ in present-day (PD) aerosol perturbation experiments as observed in non-precipitating stratocumulus clouds. However, the manuscript quickly shows that these negative correlations are not causal. The models still simulate a positive relationship between simulated LWP and $N_d$ (likely through precipitation suppression). The remainder of the study provides further evidence that the GCM diagnosed relationship is indeed non causal and exemplifies how so-called confounding factors (in this case precipitation or PBL-depth), or simply statistical sampling and parameter-space limitations. The authors argue, that such confounding factors may not only be limited to GCMs, but may also impact observation-based PD climatologically diagnosed relationships between LWP and $N_d$. And it concludes with a general need in the community to find joint avenues forward to disentangle correlation and causation between these to cloud properties in order to reduce uncertainty of the effective radiative forcing contribution from LWP adjustments.

The study raises some very important conundrums in understanding and quantifying aerosol-cloud interactions for the scientific community. It also very clearly shows that capturing the -ve slope between LWP and $N_d$ in GCMs alone is insufficient to increase our confidence in the LWP adjustment and its correct mechanistic implementation in GCMs (which often miss the proposed causal mechanism altogether). The manuscript is very well written and follows a nice and clear story line. I thus recommend publication following minor revisions.

Comment on Methodology:

My most general comment is with respect to the statistics used in this study in Figs 7. and 11., which are discussed in sections 3.3.1 and 3.3.2. You introduce two distinct confounding variables here in these sections: surface precipitation rate or boundary layer depth. Both of these variables, as you state are not independent from your predictor variable $N_d$ (and indeed your response variable LWP). The problem in binning in one variable, say PBL depth, and then looking at the slope in linear log space between averaged $N_d$ and LWP is that you are already averaging out some of the co-variability that undoubtedly exists between predictor and response variable in each PBL depth bin. It thus skews your statistic (unless you got lucky) and the slope of the linear regression you obtain. It would be more accurate to assume that your response LWP variable co-varies with Nd and PBL and do a multi-variate fit. Or said differently: if you have an expression LWP= const. $H_{pbl}^a N_d^b$, then you can determine "a" and "b" using partial derivatives in log space. Note though that when integrated, these are only valid up to a constant! Therefore when determining a:=dln(LWP)/dln($N_d$) at constant $H_{pbl}$, don't average, but fit slope instead.

Minor Comments:

- Can you provide a solid argument for the 30% occurrence threshold. if not, how sensitive are your results to that parameter choice?

- L212: PBL depth only goverened by anticyclonic subsidence? What about the gradient in SST?

- Figs 8 and following: model level is not a meaningful quantity for people not directly involved in the study. Please provide more meaningful height intervals.

Edits:

- L53: I would remove brackets, its a stand-alone sentence
- L75: Please state explicitly that all other experiments use model diagnosed LWP and $N_d$.
- Figs 3 and following: Are these normalised PDFs around the edges? I don't remember seeing this written anywhere.
- Fig4 caption: I would include info that its CMIP6 era experiments in caption
- Fig7: clarify that rain intervals intervals are given in brackets
- L184: sentence containing „$N_d$ distribution is noticably lower" is ambiguous to me. You mean the peak in the distribution is situated at lower $N_d$? All the distributions overlap, so how do you quantify „noticeably"?
- L232: Please rephrase „... equally accessible to clouds". LWP and $N_d$ are cloud properties, so how can they not be accessible to a real cloud? You mean accessible to an observed or a simulated cloud, upon which limiters are imposed? Or do you just want to point out that the LWP and $N_d$ phase space is not populated uniformly at equal density? Please clarify.

---

## Author Comment (AC1)

**Response to RC1 (Jianhao Zhang)**

*Thank you for your thoughtful review and suggestions for improved presentation. We enjoyed engaging with your ideas and questions for future work. Our responses are inline below.*

**Stimulated thoughts after reading (rather than comments):**

- Regarding the disagreement between the causal experiment (PD-PI) and the internal variability, is there a possibility that the causal experiment is missing some feedback mechanisms (at longer timescales) that may be present in the internal variability, because of the fixed climatic boundary conditions? (I realize this may not contribute much to the disagreement, but just wondering. . . ) About the climatic boundary conditions, you mentioned that SST is fixed, is the circulation (winds) also fixed? (I'm not very familiar with the setup of these experiments) For now, let's assume the internal variability (inverted v) captures the mean climate state where MET (large-scale conditions), Nd, and LWP are in balance (manifested in some climate scale correlations), perturbing Nd initially causes changes in LWP, which may later lead to circulation and/or SST changes (feedback from LWP to MET, and then possibly back to LWP). Is this potential feedback pathway artificially shutoff in these PI-versus-PD runs, based on the configuration?
  *It is quite plausible that these mechanisms contribute in the real climate system. To include these effects in our model experiments, we would need to remove the fixed sea surface temperature and the nudging to the circulation. This would be more expensive, because it would require running the ocean model and would require longer integration time to average out the atmosphere/ocean coupled modes of variability; it would also mix in a cloud feedback signal because the aerosol ERF would cause the SST to decrease in a coupled experiment. Nevertheless, we agree – we are trying to understand a multiscale system, and artificially removing the circulation-mediated part of the response can only be an interim solution. We have added a caveat to the manuscript.*

- Regarding the "funny" "doubly surprising" thing happened in CMIP6 models. I'm just curious is there any clue on what causes the CMIP6 models to get this inverted v (I understand the case for ModelE)? Are there any speculations? Is this due to the fact that the newer version of models better capture the mean climate states, thereby closer to observationally derived correlations? A following question is that if you use AeroCom IND3 models to predict the PI-LWP, would you get agreement with the causal experiment?
  *This is still a puzzle. Unfortunately, the intersection between CMIP5 AeroCom models and CMIP6 models in this study is small (only CAM5/CAM6). We hope that an updated AeroCom experiment will provide comparisons between the CMIP5 and CMIP6 versions of more models. We also hope perturbed physics ensembles of each of the "inverted v" models will explore the effect of physics choices on the $N_d$–$\mathcal{L}$ relationship. We have included this response in the revised manuscript, since many other readers may have this question, too.*

**Some notes:**

- Line 77, check spelling "ObservaTon"
  *Corrected, thanks!*

- Figure 7-12, perhaps it's worth mentioning these are results from E3SM in the captions? (I know this is clearly indicated in the main text, so, feel free to ignore this).
  *Thanks for suggesting this addition. We have made this change to the figure captions.*

- Figure 9, wind vectors are kind of hard to see, I suggest enlarging them (perhaps fewer of them will help too); is it better to indicate translated PBL depth in pressure or meter (more intuitive units)?
  *Thanks for the suggestions. We have made the wind vectors fewer and larger. We have noted the (mean ± standard deviation, since the model uses hybrid sigma coordinates) PBL geometric depths in all figures that make reference to the PBL depth.*

- Line 229 & Figure 11, regarding Nd-LWP correlation within each PBL depth bin (not shown), I wonder if it's worth showing, as I am curious about whether they look similar to what have been shown in Figure 7, i.e., in classic Simpson fashion, or different?
  *We have decided against plotting the stratified regressions, which lie on top of each other and make for a messy plot. We have instead noted the very narrow range of regression slopes (i.e., not Simpson-like behavior) in the text.*

- Just want to say that I really enjoyed reading Section 3.4 and the conclusion part. Great discussions! and I think the ACI community should really think carefully along these lines (i.e., representativeness versus/and causality) before producing tons of papers on the topic while not sure about how much of the results are causal.
  *Thank you! Fortunately, most of the community recognizes these issues as caveats. We hope this study will provide further impetus to address the representativeness and causality problems.*

**Response to RC2 (Anna Possner)**

*Thank you for the thoughtful review and valuable suggestions for improving the analysis of stratified $N_d$–$\mathcal{L}$ relationships and the presentation. Our responses are inline below.*

**Comment on Methodology:**

My most general comment is with respect to the statistics used in this study in Figs 7. and 11., which are discussed in sections 3.3.1 and 3.3.2. You introduce two distinct confounding variables here in these sections: surface precipitation rate or boundary layer depth. Both of these variables, as you state are not independent from your predictor variable Nd (and indeed your response variable LWP). The problem in binning in one variable, say PBL depth, and then looking at the slope in linear log space between averaged Nd and LWP is that you are already averaging out some of the co-variability that undoubtedly exists between predictor and response variable in each PBL depth bin. It thus skews your statistic (unless you got lucky) and the slope of the linear regression you obtain. It would be more accurate to assume that your response LWP variable co-varies with Nd and PBL and do a multi-variate fit. Or said differently: if you have an expression LWP = const. $H_{\text{pbl}}^a$ $Nd^b$, then you can determine $a$ and $b$ using partial derivatives in log space. Note though that when integrated, these are only valid up to a constant! Therefore when determining $a$:=dln(LWP)/dln(Nd) at constant $H_{\text{pbl}}$, don't average, but fit slope instead.
*Thank you for this suggestion. We have added monovariate and bivariate linear regressions for the potential confounding by precipitation and PBL depth to the text. The bivariate regression results are consistent with the binned analysis, but we agree that they provide useful quantitative information as well.*

**Minor Comments:**

- Can you provide a solid argument for the 30% occurrence threshold. if not, how sensitive are your results to that parameter choice?
  *We have updated the text and redesigned Fig. 1 to illustrate the rationale behind the $f_{Sc}$ threshold. The aim was to select the subtropical Sc regions in a self-consistent way (i.e., recognizing that the model's Sc regions might be shifted with respect to observations). Thus, we chose a round-number contour that consistently excludes the midlatitudes across the globe; the limiting factor is the northward extent of the NEP Sc region. Setting the threshold at $f_{Sc} > 0.2$ would also have been a reasonable choice. As shown in Fig. R1, the $N_d$–$\mathcal{L}$ relationship in the $0.2 < f_{Sc} \leq 0.3$ bin looks similar to the $f_{Sc} > 0.3$ relationship, so this alternative threshold choice would not have greatly changed the $N_d$–$\mathcal{L}$ relationship.*

- L212: PBL depth only goverened by anticyclonic subsidence? What about the gradient in SST?
  *The SST influence is undoubtedly the major contributor on long timescales, and this is acknowledged in the discussion of the spatial covariability of cloud and aerosol properties immediately preceding this sentence.*

*Here, the concern is spatial covariability at a fixed location; our assumption is that the subtropical anticyclone, and thus the location of the subsidence maximum and continental aerosol advection on synoptic timescales, is a stronger contributor than the slowly varying SST field. We have clarified in the text that we are referring to synoptic timescales and that an analysis across timescales would be useful.*

- Figs 8 and following: model level is not a meaningful quantity for people not directly involved in the study. Please provide more meaningful height intervals.
  *Thanks for pointing this out. We have added the mean and standard deviation geometric PBL depths to the figures.*

**Edits:**

- L53: I would remove brackets, its a stand-alone sentence
  *Agreed, thanks!*

- L75: Please state explicitly that all other experiments use model diagnosed LWP and Nd.
  *We have adopted this suggestion.*

- Figs 3 and following: Are these normalised PDFs around the edges? I don't remember seeing this written anywhere.
  *We have made the wording of the captions more precise to specify that the marginal distributions are probability distributions and that the $N_d$–$\mathcal{L}$ relationships depicted are conditional probability distributions.*

- Fig4 caption: I would include info that its CMIP6 era experiments in caption
  *Agreed, thanks for this suggestion.*

- Fig7: clarify that rain intervals intervals are given in brackets
  *We have adopted this suggestion.*

- L184: sentence containing „Nd distribution is noticably lower" is ambiguous to me. You mean the peak in the distribution is situated at lower Nd? All the distributions overlap, so how do you quantify „noticeably"?
  *Thanks for noting this ambiguity. We have changed the text to "the peak of the $N_d$ distribution is shifted lower".*

- L232: Please rephrase „... equally accessible to clouds". LWP and Nd are cloud properties, so how can they not be accessible to a real cloud? You mean accessible to an observed or a simulated cloud, upon which limiters are imposed? Or do you just want to point out that the LWP and Nd phase space is not populated uniformly at equal density? Please clarify.
  *Thanks for this comment. We did mean, as you surmised, that "not all parts of the $N_d$–$\mathcal{L}$ phase space are uniformly populated by clouds" and have changed the text to that phrasing.*

[Figure]

Figure R1: Variation of the $N_d$–$\mathcal{L}$ relationship with $f_{\mathrm{Sc}}$. The $0.3 < f_{\mathrm{Sc}} \leq 1$ bin corresponds to the selection used in the manuscript.